# Novel Thermally Reduced Graphene Oxide Microsupercapacitor Fabricated via Mask—Free AxiDraw Direct Writing

**DOI:** 10.3390/nano11081909

**Published:** 2021-07-25

**Authors:** Vusani M. Maphiri, Gift Rutavi, Ndeye F. Sylla, Saheed A. Adewinbi, Oladepo Fasakin, Ncholu Manyala

**Affiliations:** 1Department of Physics, Institute of Applied Materials, SARChI Chair in Carbon Technology and Materials, University of Pretoria, Pretoria 0028, South Africa; vusanimuswamaphiri@gmail.com (V.M.M.); rutavigift@yahoo.com (G.R.); ntoufasylla@gmail.com (N.F.S.); fasakinoladepo@gmail.com (O.F.); 2Department of Physics, Osun State University, Osogbo, Osun State 210001, Nigeria; adeksaheed007@gmail.com

**Keywords:** microsupercapacitor, direct writing, energy storage, graphene oxide, thermally reduced

## Abstract

We demonstrate a simple method to fabricate all solid state, thermally reduced graphene oxide (TRGO) microsupercapacitors (µ-SCs) prepared using the atmospheric pressure chemical vapor deposition (APCVD) and a mask-free axiDraw sketching apparatus. The Fourier transform infrared spectroscopy (FTIR) shows the extermination of oxygen functional groups as the reducing temperature (RT) increases, while the Raman shows the presence of the defect and graphitic peaks. The electrochemical performance of the µ-SCs showed cyclic voltammetry (CV) potential window of 0–0.8 V at various scan rates of 5–1000 mVs^−1^ with a rectangular shape, depicting characteristics of electric double layer capacitor (EDLC) behavior. The µ-SC with 14 cm^−2^ (number of digits per unit area) showed a 46% increment in capacitance from that of 6 cm^−2^, which is also higher than the µ-SCs with 22 and 26 cm^−2^. The TRGO-500 exhibits volumetric energy and power density of 14.61 mW h cm^−3^ and 142.67 mW cm^−3^, respectively. The electrochemical impedance spectroscopy (EIS) showed the decrease in the equivalent series resistance (ESR) as a function of RT due to reduction of the resistive functional groups present in the sample. Bode plot showed a phase angel of −85° for the TRGO-500 µ-SC device. The electrochemical performance of the µ-SC devices can be tuned by varying the RT, number of digits per unity area, and connection configuration (parallel or series).

## 1. Introduction

Carbon is a naturally abundant element, which is also very interesting due to its ability to self-bind in different ways giving rise to various carbon structural allotropes such as graphene, carbon nanotubes, diamond, and fullerenes [1,2,3,4]. Some of the allotropes have attracted great attention due to their nanoscale, surface area, shape, and peculiar dimensions. Graphene is a two-dimensional single layer of carbon atoms with astonishing electrical, thermal, and mechanical properties. This is because atoms are bonded through sp^2^ hybridization, giving rise to the three in-plane σ bonds reposible for flexibility and strength. The weak, out-of-plane π bond is responsible for electrical and thermal properties [5,6]. The presence of these bondings makes graphene one of the best materials suitable for microsupercapacitor (µ-SC) application [6,7]. However, the abovementioned properties are only observed in defect-free graphene, which is difficult and expensive to mass produce. Alternatively, there are cheaper and more simple ways to mass produce a slightly defective graphene, such as atmospheric pressure chemical vapor deposition (AP-CVD) [8,9]. The chemical phase exfoliation via the Hummer’s method of graphite oxide has been widely used, but this method produces a highly oxidized version of graphene [5,10,11,12,13,14,15,16,17,18].

Several techniques have been used to reduce oxygen on GO, such as the use of chemicals, UV radiation, and electrochemical and thermal methods [6,11,18,19]. Thermal reduction of graphene oxide is a quick, cheap, safe, and relatively easy method and has never been reported on the fabrication of µ-SC in this manner. On the contrary, thermal reduction has been reported for supercapacitor application by Zhao et al. [13] where the reducing temperature (RT) was varied from 200 to 900 °C in a nitrogen environment for two hours. The Fourier transform infrared spectroscopy (FTIR) showed the decrease in notches attributed to the various oxygen related vibrations as a function of the RT. This behavior was attributed to the removal of oxygen functional groups and the intercalated H_2_O. The highest capacitance of 260.5 F g^−1^ at 0.4 A g^−1^ was obtained at the RT of 200 °C, and 11.7% (30.6 Fg^−1^ at 0.4 Ag^−1^) of the initial capacitance was retained at a RT of 900 °C. This decrease was also attributed to the removal of the oxygen functional groups which served as an ion passage to the internal surface. Like supercapacitors, µ-SC are new µ-energy-storage units with similar energy storage mechanisms but are smaller in size. This expands the already huge potential applications of supercapacitors into the realm of small and high power density devices, with practical applications in medical implants, wearable technology, and many more [20,21,22]. These devices outperform other µ-energy-storage devices like micro-batteries because of ultra-fast charge and discharge rates, long cycling lifetime, cost efficiency, and operational safety and efficiency [2].

µ-SCs have been fabricated and patterned via several techniques classified as masked and mask-free techniques. The masked techniques use a photoresist such as Poly (methyl methacrylate) (PMMA) which can be removed by acetone, but these possess a problem of photoresistant impurities on the active material. However, mask-free is a better alternative since it does not require any photoresist. There are several mask-free techniques such as micro-plasma-jet, axiDraw, and laser reducing [2,23,24,25,26]. These methods are integrated into a computer driven stage control which is very different to the masked technique. Liu et al. [2] used mask-free ambient micro-plasma-jet etching to pattern µ-SC on a multi-walled carbon nanotube (MWNT) on a Polyethylene terephthalate (PET) substrate. A solid-state polyvinyl alcohol-H_3_PO_4_ gel electrolyte was used and a stack capacitance of 2.02 F cm^−3^ was obtained at a scan rate of 10 mV s^−1^. Park et al. [26] reported a laser-patterned stretchable µ-SC for soft electronic circuit components using Polydimethysiloxane as a substrate. The reduced graphene oxide/gold composite active material showed a higher volumetric energy density of 290 µWh cm^−3^. In this study, we used the axiDraw, similar to Ref [23,24] who reported the pattern MXene-based material for µ-SC application. This is because axiDraw does not involve expensive patterning accessories such as laser and plasma jet equipment. The axiDraw was equipped with an inexpensive sharp blade pen that requires sharpening and can have a long lifespan compared to the laser and helium and oxygen cylinder for the plasma jet.

This work demonstrates a novel µ-SC fabricating method, where GO films are thermally reduced directly on the microscopic glass (MSG) substrate followed by a patterning via a mask-free axiDraw apparatus. In comparison with other µ-SC preparation techniques [24,25], thermal reduction combined with axiDraw direct writing is a relatively quick, easily scalable, and safe method for production of µ-SCs. Mao et al. [25] reported an all-solid-state flexible microsupercapacitors (µ-SCs) based on reduced graphene oxide/multi-walled carbon nanotube (MWCNT) composite; the GO/MWCNT was sprayed on PET and dried for 24 h and followed by the laser reduction process for 30 min. The µ-SC showed a volumetric energy density of 6.47 mW h cm^−3^ and a cycling stability retention of 88.6% after 10,000 cycles. Li et al. [24] reported a MXene-conducting polymer electrochromic microsupercapacitor prepared by electrochemical deposition of poly(3,4-ethylenedioxthiophene) on Ti_3_C_2_T_x_ MXene. The as prepared PEDOT/ Ti_3_C_2_T_x_ was washed and dried for 6 h and showed an areal capacitance of 2.4 mF cm^−2^ at 100 nm thickness. These methods, reported in references [24,25], took more time as compared to the proposed method which yielded a volumetric energy density of 14.61 mW cm^−3^ and volumetric power density of 142.67 mW h cm^−3^ for TRGO-500. This high RT yielded µ-SCs with rectangular cyclic voltammetry (CV) curves with high scan rate, isosceles triangle galvanostatic charge-discharge (GCD) curves, higher capacitance retention, low equivalent series resistance (ESR), high phase angle and diffusion length closer to the imaginary impendence axes.

## 2. Experimental

### 2.1. Materials

Chemicals used in this article were analytical grade and were used without any further purification. The used material and chemicals are as follows: Natural graphite, sulphuric acid (H_2_SO_4_ (98%)) (associated chemical enterprises, Johannesburg, South Africa ), ortho-phosphoric acid (H_3_PO_4_) (MERCK, Darmstadt, Germany), microscopic glass (MSG) (B&C Glass Ltd., Haverhill, UK), potassium permanganate (KMnO_4_) (associated chemical enterprises, Johannesburg, South Africa), hydrogen peroxide (H_2_O_2_ (50%)) (associated chemical enterprises, Johannesburg, South Africa), poly (Vinyl Alcohol) (PVA, Mw 89,000–98,000) (Sigma-Aldrich, Steinheim, Germany), ethanol (C_2_H_5_OH) (associated chemical enterprises, Johannesburg, South Africa)and deionized water (H_2_O) (DW) (prepared using DRAWELL, laboratory water purification system) at 18.2 MΩ.

### 2.2. Preparation of GO

GO was prepared using the modified Hummer’s method [10]. Graphite powder was slowly added into a cooled H_2_SO_4_ and stirred until it reached homogeneity. KMnO_4_ was then added into the cooled homogeneous solution. The solution was stirred for 180 min at a constant temperature of 50 °C and then left to cool to room temperature. Then 100 mL of H_2_O was added followed by 20 mL of H_2_O_2_ to stop the reaction by reducing the remaining KMnO_4_ into a water-soluble Manganese sulphate (MnSO_4_), as described by Equation (1) below:(1)$$2{\mathrm{KMnO}}_{4}+3H_{2}{\mathrm{SO}}_{4}+5H_{2}O_{2}=2{\mathrm{MnSO}}_{4}+K_{2}{\mathrm{SO}}_{4}+8H_{2}O+5O_{2}$$

The GO was cleaned by adding DW into the GO solution and letting it settle, then decanting the DW. This process was repeated several times to ensure that GO was free of impurities. The GO mixture was then centrifuged at 5000 rpm for 5 min. Obtained GO subsequently dried in an oven at 60 °C and was stored for µ-SC fabrication and characterization.

### 2.3. Preparation of Microscopic Glass (MSG)

The MSG was cleaned with acetonitrile soap to remove any greasy substances, then rinsed with DW. The MSG was then ultrasonicated for 30 min in a piranha solution with a ratio of 1:4 of H_2_SO_4_ to H_2_O_2_. The MSG was then rinsed with DW and blown dry using N_2_ gas.

### 2.4. Preparation of TRGO Film and µ-SC Assembly 

In total, 5 mg of dried GO was added into 1 mL of ethanol and the mixture was sonicated for 30 min. The GO/ethanol solution was loaded into the spay gun (MASTER AIRBRUSH, Model G233, San Diego, CA, USA) and spray coated onto MSG. The GO film on the microscopic glass was thermally reduced using atmospheric pressure chemical vapour deposition (APCVD) at various temperatures from 100–500 °C for 10 min at a heating rate of 10 °C min^−1^ in the presence of argon (Ar = 200 sccm) and hydrogen (H_2_ = 20 sccm). The samples were labelled as TRGO-100, TRGO-200, TRGO-300 TRGO-400, and TRGO-500; and the numerical value at the end of “TRGO-” signifies the RT. At temperatures higher than 500 °C the MSG melted/deformed, and the material also peeled off the surface of the MSG. axiDraw (Model V3, Evil Mad Science LLC, Sunnyvale, CA, USA) controlled via Inkscape (0.92.4, Boston, MA, USA) using the extension axiDraw Control version 2.5.3 was used as an X-Y stage controller for the µ-SC pattern. A sharp blade pen attached to the axiDraw was used to write µ-SC patterns on the thermally reduced graphene oxide (TRGO) film. This process is schematically depicted in Figure 1. The unwanted TRGO was scraped off the MSG. Copper strips were placed on to the edges of the µ-SC and adhered using a kapton tape. The digital images of the spray gun, GO thin film on microscopic glass, axiDraw, sharp blade pen, and TRGO film patterned µ-SC with 6, 14, 22, and 26 digits on microscopic glass are displayed in the Appendix A (Appendix A).

### 2.5. Preparation of PVA-H_3_PO_4_ Gel Electrolyte

Proton conducting polymeric electrolyte (PVA-H_3_PO_4_) was prepared by adding 1 g PVA powder into 10 mL DW. This mixture was heated at 90 °C under constant stirring until the solution turned from milky white into a clear viscous solution. The prepared PVA was placed in an oven at 60 °C to remove excess water; 30% (in volume) of H_3_PO_4_ was then added into the PVA gel and stirred for 30 min. The gel electrolyte was then drop-casted onto the µ-SC digits.

### 2.6. Compositional, Morphological, and Structural Characterization

The TRGO samples, together with precursor graphite, intermitted with GO and MSG substrate were analyzed using the following techniques: Fourier transformation infrared (FT-IR) spectroscopy was performed using the Bruker Alpha platinum-ATR (Billerica, MA, USA) (attenuated total reflection) in the range of 4000 to 400 cm^−1^ to effectively study the functional groups. Confocal WITec alpha (Ulm, Germany) 300RAS+ Raman microscopy was used to investigate the graphitic structure with 532 nm excitation laser at room temperature. The laser power was set to lesser than 2 mW to avoid heating the sample. The surface morphology of graphite and the GO and TRGO films were analyzed using a Zeiss Ultra-plus 55 field emission scanning electron microscopy (FE-SEM) (Akishima-shi, Japan) at 2 kV for high resolution.

### 2.7. Electrochemical Characterization

The electrochemical analysis (cyclic voltammetry (CV), galvanostatic charge-discharge (GCD), and stability) and electrochemical impendence spectroscopy (EIS) were measured at room temperature using the Bio-Logic VMP-300 potentiostat (Knoxville, TN, USA), controlled using the EC-Lab software (V10.37, Edmonton, AB, Canada). The measurements were performed in a two-electrode configuration (or device). 

## 3. Results and Discussion

### 3.1. Vibrational Spectroscopy

The ATR-FTIR was deployed to analyze the functional groups of the prepared sample, and the obtained spectra are displayed in Figure 2a. The spectrum of graphite show the absence of any absorption bands indicating chemical inertness [12]. Upon the oxidation of graphite, the spectrum of the sprayed GO on MSG showed several absorption bands attributed to various O-functional groups. The broad band at 3314 cm^−1^ was ascribed to the stretching vibration of O-H, indicating the presence of hydroxyl (-OH) and/or carboxylic (-COOH) functional groups within the GO structure. Less intense stretching vibration of the carbonyl (C=O) and unoxidized graphitic corresponding to the (C=C) were attributed to the 1734 and 1633 cm^−1^ notches, respectively. The oxygen singly bonded to the stretching vibration of O-H deformation; C-O of the epoxy group and C-O of the alkoxy group were attributed to the 1407, 1220, and 1047 cm^−1^, respectively. Similar, FTIR spectra of GO was reported in Refs. [12,13,14]. As the GO thin film was subjected to a thermally RT of 100 °C, some O-containing groups were removed as part of the H_2_O and CO_2_ gases. This can be seen by the peak intensity decrease of the 3341, 1734, and 1407 cm^−1^ attributed to either O-H or C=O stretching vibration. The peak at 878 cm^−1^ can be attributed to the Si-O from the MSG substrate [27,28]. Above the thermal RT of 100 °C, the spectra are slightly like that of graphite, showing a low degree of oxygen function groups. Similar results were reported by Zhao et al. [13] when varying the thermal RT of powder GO: the FTIR showed the removal of O-containing functional groups as temperature increased.

Raman spectra of the prepared sample are displayed in Figure 2b. In general, the obtained spectra showed the presences of D, G, 2D, D + G, and 2D’ occurring at around 1363, 1604, 2714, 2936, and 3197 cm^−1^, respectively. The graphite only exhibited the D, G, and 2D peaks. The D peak is due to breathing mode of k-point phonons of A_1g_ symmetry (associated with the local defects and disorder of the edges of graphene and graphite materials). The G peak is due to the first-order scattering of E_2g_ phonon of sp^2^ C atoms (associated with the high-order hexagonal structure within graphite). The 2D is an overtone of the D peaks which is a fingerprint of the graphene sheet formation and number of layers [12,13,14,17,29]. Upon oxidation of the graphite, the Raman spectra of GO showed significant differences which include the broadening of phonon range peaks and the emergence of two overtone peaks (D + G and 2D′). The broadening of the D peak from the full width at half maximum (FWHM) of 34.9 to 117.1 cm^−1^ is due to the attachment of the O-containing functional groups on the graphitic sheets in the oxidation reaction process [30]. The FWHM of the G peak increasing from 20.1 to 66.9 cm^−1^ is also due to the bond angle disorder caused by the attachment of the O-containing functional groups. This causes the average ideal graphite-like hexagonal 120° bond angle to change. The FWHM of the D and G peak seems not to decrease regardless of the increase in temperature. This was also observed by Claramunt et al. [31] when thermally reducing GO in the range of 100 to 900 °C. The only noticeable difference is the fading of the overtone peaks which suggests that the layers are increasing (or restacking), since they only dominate in graphitic materials containing few layers of graphene sheets. Thus, the stacking increases as temperature increases, leading to low 2D intensity. This is also confirmed by the FTIR spectra showing the decrease of oxygen functional groups present on the graphene sheets causing them to restack into graphite-like material.

### 3.2. Scanning Electron Microscopy

The SEM micrographs of the graphite, GO, TRGO, and µ-SC (interspace distance and digit width) are displayed in Figure 3. The micrograph in Figure 3a shows particles of graphite with random particles sizes which are flat or disc in nature and are highly stacked together. This morphology has been previously reported by Dai et al. [32] and Montagna et al. [33]. The morphology in Figure 3b–d depicts a flat sheet stacked together [18]. In Figure 3b, the GO sheets are lousily stacked compared to TRGO-100 (Figure 3c) and TRGO-500 (Figure 3d). The insets on Figure 3b–d show that the GO and TRGO sheets are wrinkled. The SEM micrographs of µ-SC interspace *i* and digit width *W* are displayed in Figure 3e,f, respectively. The interspace of all the prepared µ-SC is ~38µm, which is due to the dimension of the pattering apparatus (sharp blade pen) which has a sharp tip (see Appendix A). The 1.6 mm digit width corresponds to the digit width of the µ-SC (6). The µ-SC dimensions of all fabricated µ-SC are given in Appendix A.

### 3.3. Electrochemical Results

The digital photography image of the fabricated µ-SC is displayed in Figure 4. The µ-SC is composed of only five components (also mentioned in Section 2.4, the preparation of TRGO films and µ-SC assembly): MSG as a substrate, TRGO as an active material and a current collector because of its conductivity; copper foils as positive and negative terminals; and kapton tape which adheres the copper foil onto the current collector (edge (E) of the µ-SC). The inkscape schematics of the µ-SCs are displayed in Appendix A, showing various numbers of digits per unit area (*n*) of 6, 14, 22, and 26; and other parameter such as breadth (*B)*, length *(L)* and width (*W)* of the µ-SC, respectively. The digital photograph of those µ-SC patterns on TRGO are displayed in Figure 5a. To determine the number of digits per unit area giving the best electrochemical performance, electrochemical studies were performed on the TRGO-300. The CV and GCD curves of the TRGO-300 µ-SC are displayed in Figure 5b,c, respectively. Note that the number within the parenthesis at the end of “µ-SC” denotes the number of digits per unit area (cm^−2^). The CV curves depict a rectangular shape indicating the electric double-layer capacitor behavior, while the GCD curves show a triangular shape further confirming the EDLC nature of the material. The CV and GCD curves suggest that the µ-SC (14) has the better charge carrying ability because it shows higher current response and longer discharge time than others. The areal capacitance ($C_{Areal}$) displayed in Figure 5d of the TRGO-300 µ-SC was estimated from the CV curves using Equation (2) below [2,24]:(2)$$C_{Areal}=\;\frac{\int_{\;V_{i}}^{V_{f}\;}idV}{2S\left(V_{f}-V_{i}\right)A}$$
where $V_{i}$ and $V_{f}$ are the initial and final values of the working potential, respectively. i, S, and A are the current, scan rate (20 mVs^−1^), and total surface area of the µ-SC, which is including the inter-digit space between the adjacent digits. The surface area of all the µ-SC is 1 cm^2^ (see Appendix A, Appendix A). The areal capacitance versus the number of digits per unit area is illustrated in Figure 5d which shows that the best areal capacitance occurred at 14 cm^−2^. The increase of the areal capacitance up to 14 cm^−2^ can be attributed to the reduction of the average ionic diffusion pathway between adjacent digits [2,34]. This was shown by Liu et al. [2] where the µ-SC with 12 digits had more capacitance than those with four and eight digits. This increase can be also explained by the distributed capacitance effect in Ref. [20] which suggests that higher digits lead to higher electrochemical performance. From previous studies, it seems that more digits per unit area results in a better electrochemical performance, which includes more power and energy. This seems to be true for a certain range of digits per unit area due to external factors such as the removal of active electrode mass which increases as the number of digits increase [35] (see Appendix A, Appendix A) and the electric field strength which increases as the number of digits increases due to the edging effect [21]. The areal capacitance of the TRGO-300 determined from the total surface area (area of the entire µ-SC including the interspace) and effective area (excluding the interspace) is displayed in Appendix A. The effective area and total interspace area were calculated using the derived Equations (S1) and (S2), respectively. It is clear that the removed material while pattering accounts only for a small fraction of the loss of capacitance [20,21]. Thus, the increase and decrease of capacitance can be attributed to the increase of the electric field strength around the edges of the interdigitated electrode. It is well known that the electric field line strength increases for edge intensive electrodes rather than a continuous or round shape electrode [36]. This was illustrated by Hota et al. [21] in a simulation comparing the electric field generated between the interdigitated and fractal electrode. The Moore fractal electrode design showed a 32% increase in energy density over the conversational interdigitated electrode. This is because the Moore fractal is more edge intensive than the interdigitated electrode design. This behavior suggests that increasing the number of digits per unit area subsequently increases the number of edges leading to a high electric field distributed along the edges within the µ-SC. In this work, the areal capacitance of the 14 cm^−2^ increased by 46% over the capacitance of 6 cm^−2^. Moreover, a very strong electric field concentrated within the same region (between the end and edge of the µ-SC, Appendix A) separated by a narrow interspace (38 µm, Figure 3f) leading to charge flow leakage (short circuit) between electrolyte and electrode led to a decrease in capacitance [21,37]. Thus, 14 cm^−2^ is the optimum electrode design due to an adequate ratio of interdigitated electrodes per unit area and electrode width interspace ratio [37,38,39] that maximizes the electric field strength while avoiding electric charge leakage and manages to reduce the ionic diffusion pathway.

Since 14 cm^−2^ µ-SC configurations gave the best electrochemical results as compared to the other digit configurations, it was used throughout the entire study to analyze the effect of thermal reduction on the µ-SC electrochemical performance. The CV curve measurements at 20 mV s^−1^ of the µ-SCs at different RTs are displayed in Figure 6a. Note that the CV of TRGO-100 was not measured due to its high oxygen functional groups (see the FTIR in Figure 2a) present within the sample. All the CV curves at various RTs were measured in the voltage window ranging from 0 to 0.8 V. Figure 6a shows that the response in current density (mA cm^−2^) decreases as the RT increases due to the decrease of the oxygen functional groups on the surface of the graphene sheet. This is because these functional groups play a very important role as they serve as a passage for the ions into the internal surface [13] and also provide wettability between the electrolyte and active material [40]. Figure 6b shows the areal capacitance (displayed in Appendix A) at various scan rates at different temperatures. The TRGO-200 µ-SC has a higher areal capacitance of 0.5421 mF cm^−2^ at 10 mV s^−1^ and retains 27% (0.1498 mF cm^−2^) at a scan rate of 80 mV s^−1^. TRGO-300 µ-SC has an initial areal capacitance of 0.1227 mF cm^−2^ at 10 mVs^−1^ while retaining 41% (0.0506 mF cm^−2^) at 200 mV s^−1^, and TRGO-400 and TRGO-500 have areal capacitances of 0.0382 and 0.0298 mF cm^−2^ at 10 mV s^−1^ while retaining 36% (0.0140 mF cm^−2^) and 53% (0.0159 mF cm^−2^) at 1 V s^−1^, respectively. Despite the low areal capacitance, TRGO-500 has a better retention and shape. 

The Nyquist plots, showing imaginary (ImZ) and real (ReZ) impedances at various frequencies of different RTs, are displayed in Figure 6c, with an inset showing a zoomed view of the high-frequency region. This behavior of the EIS (Nyquist plot) was suggested by Mathis et al. [41] as an ideal SC behavior [42], when a 45° line starting from the equivalent series resistance (ESR) quickly transcends into a vertical line that is parallel or close to parallel to the ImZ axis at the low-frequency region. This also means that the reactive sites leading to a capacitor-like behavior are fully accessible in a short time. The absence of a semi-circle at the high-frequency region shows the absence of the charge transfer. The µ-SCs have equivalent series (or solution) resistances (ESR) of 4705, 3595, 2584, and 2326 Ω for TRGO-200, TRGO-300, TRGO-400, and TRGO-500, respectively. The ESR as a function of RT is displayed in Appendix A, and it can be seen that the ESR reduces as RT increases, suggesting that oxygen has a great influence on the ESR. The TRGO-500 shows a vertical line in the low-frequency region which is more closer to the Im Z (*Y*-axis) axis indicating a better capacitive behavior evident that all the reactive sites are fully accessible in a short time [41,43] due to the increase of higher carbon content (see Figure 2a). This is verified by the phase angle (as a function of frequency) shown in the bode plot in Figure 6d. The phase angle in the low-frequency region shows an increase from −48 to −85° as the RT increases from 200 to 500 °C. At 500 °C, the obtained phase angle is very close to the ideal value of −90 which suggests an ideal capacitive behavior [41,42]. This was also seen on the normalized CV curves displayed in Figure 6b where the TRGO-500 µ-SC has the better rectangular shape than those at low RTs. The normalized capacitance C″ versus frequency is illustrated in the insert. The relaxation time ($\tau_{O}$) which is the transition point of the electrochemical capacitor from capacitive to resistive behavior—which also corresponds to the point of maximum energy dissipation—shows a decrease as the RT increases. The estimated relaxation time constant $\tau_{O}$ (reciprocal of the frequency $f_{O}$) for TRGO-300, TRGO-400, and TRGO-500 was approximately 13.6, 1.9, and 1.0 s, respectively. The TRGO-500 has a smallest relaxation time, confirming that indeed all the reactive sites are fully accessible within a short period of time. 

The CV curve of the TRGO-500 at various scan rates is displayed in Figure 7a, while those of low temperature are displayed in Appendix A. The TRGO-500 µ-SC has a high sweeping scan rate tolerance over those µ-SC prepared at low temperatures. The rectangular shape is still maintained at a higher scan rate of 1 Vs^−1^ showing a better capacitive device. The GCD curves of TRGO-500 at different areal current are displayed in Figure 7b and those of low temperature are displayed in Appendix A. All the GCD curves show a triangular shape, signifying the EDLC behavior of the µ-SC devices. The areal capacitances from the GCD curve were estimated using Equation (S3) and are given in Appendix A. The decrease of the capacitance can still be attributed to the removal of the oxygen functional groups. Figure 7b and Appendix A also show that high reduced temperature µ-SC (300 to 500 °C) has a wide current density range and can be charged and discharged by small current density and still manage to maintain a triangular shape, similar to the CV curves in Figure 7a and Appendix A. The areal and volumetric capacitances calculated from Equation (S3) are displayed in Figure 7c,d, respectively. Maximum areal and volumetric capacitances of 0.0387 mF cm^−2^ and 14.8981 mF cm^−3^ were observed for the TRGO-500 µ-SC and the thickness was calculated from the SEM image displayed in Appendix A. Both areal and volumetric capacitance decreased with increasing current density, having a rate capability of 87.2% at 1.66 µA cm^−2^.

The areal/volumetric energy ($E_{density}$) and power ($P_{density}$) densities calculated from the Equations (3) and (4) below:(3)$$E_{density}=\;\frac{1}{3600\Gamma }\int iVdt$$
(4)$$P_{density}=\;3600\times \frac{E_{density}}{\Delta t}$$
where Γ, ∫iVdt, and Δt are the area or volume of the µ-SC, the integral of the discharge curve, and discharge time, respectively. The TRGO-500 µ-SC produced a volumetric energy density of 14.61 mW cm^−3^ and volumetric power density of 140.89 mW h cm^−2^ at 0.0083 µA cm^−2^ while producing volumetric energy density of 0.6449 mW cm^−2^ and volumetric power density of 142.67 mW h cm^−2^ at 1.66 µA cm^−2^. The Ragone plot of the TRGO-500 is displayed in Figure 8a compared to similar devices in the literature. The TRGO-500 µ-SC has volumetric energy and power densities similar to those reported in Refs. [24,26,44], including lithium film batteries. This simple method produces a µ-SC device with similar energy and power to those devices prepared via a complex and sophisticated method. Note that the Ragone comparing different RTs is displayed in Appendix A. The TRGO-500 µ-SC displayed in Figure 8b shows a capacitance retention and columbic efficiency of 95% and 100%, respectively, at a current density of 0.83 µA cm^−2^ for 4000 cycles. The adaptability of the TRGO-500 µ-SC was demonstrated by fabricating two µ-SC devices together in parallel and in series. The digital photographs together with the inkscape schematic diagram of the two µ-SCs in parallel and series are displayed in Appendix A. The CV curves at 20 mV s^−1^ and GCD curves at 0.16 µA cm^−2^ are presented in Figure 8c,d, respectively. The voltage window for the single and parallel devices ranges from 0 to 0.8 V and that in series ranges from 0 to 1.6 V. The CV curves still show the rectangular shapes which imply fast ion diffusion. The GCD curve of the µ-SC in series has a 1.6 V charge/discharge at a similar time range as the single µ-SC, signifying good capacitive behavior at minimal internal resistance. In contrast, parallel µ-SC discharge time increased twice at the same current density as the single µ-SC. The areal energy density of the µ-SC connected in series is 0.0392 mW h cm^−2^ and parallel is 0.0402 mW h cm^−2^, which is approximately twice that of a single (0.0191 mW h cm^−2^) device. The areal power density of the µ-SC in series is 0.7650 mW cm^−2^, which is approximately twice that of single (0.3762 mW cm^−2^) and parallel connected devices (0.3825 mW cm^−2^).

## 4. Conclusions

In summary, we successfully demonstrated the possibility of fabricating thermally reduced µ-SC via the novel GO film reduction to TRGO film via AP-CVD combined with a musk-free direct writing via axiDraw. The FTIR showed a decrease in functional oxygen as the RT increased. Raman spectrum shows the fading of the overtone peaks due to restacking. The 14-digit cm^−2^ was found to show better electrochemical properties than those µ-SCs with 4-, 22-, and 26-digits cm^−2^; this behavior is attributed to the increase of electric field leading to an increase and decrease of capacitance. The TRGO-500 µ-SC showed to be the better performing device amongst µ-SCs reduced at low temperature, regardless of the small capacitance due to lack of oxygen functional groups that serve as pathways into the bulk material/wettability. The increase of thermally reducing temperature with µ-SC yielded a better EDLC behavior and rate capability due to the presence of low oxygen. The TRGO-500 µ-SC had the smallest relaxation time constant $\tau_{O}$ of 1s, higher phase angle of −85°, diffusion length close to the imaginary impendence, higher CV scan rate and GCD current density range, and a CV rate capability of 56% at 1 V s^−1^, indicating exceptional capacitive properties. The ability to vary RT, the number of digits per unit area, and the integration of µ-SC in a series or parallel configuration gives the possibility of controlling energy and power performance to meet different requirements of the miniaturized devices. 

## Figures and Tables

**Figure 1 nanomaterials-11-01909-f001:**
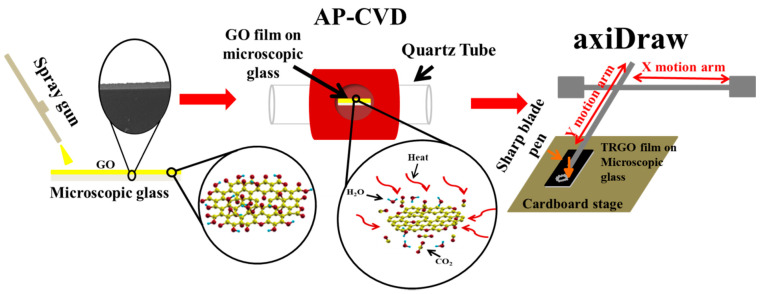
Schematic showing the TRGO µ-SC fabrication process. Spray coating prepared GO film on the MSG, thermally reducing GO film into TRGO film. Direct writing interdigitated patterns on TRGO film was completed using axiDraw for µ-SC electrodes.

**Figure 2 nanomaterials-11-01909-f002:**
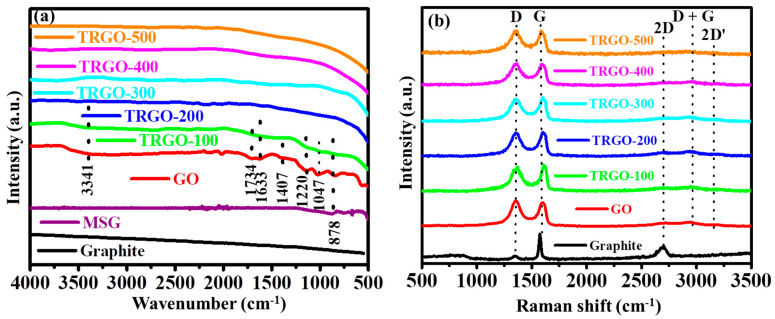
(**a**) ATR-FTIR and (**b**) Raman spectra for graphite, MSG, GO, and TRGO-100 to TRGO-500.

**Figure 3 nanomaterials-11-01909-f003:**
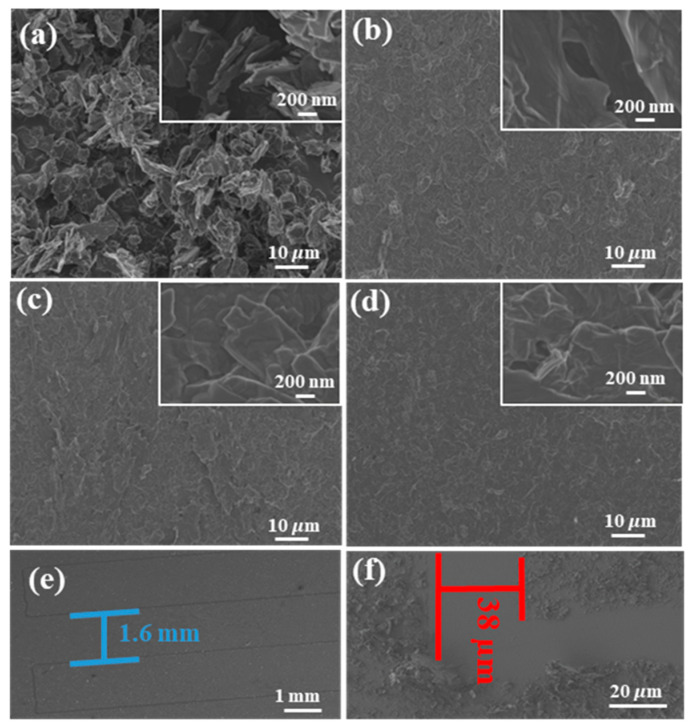
SEM micrograph of (**a**) graphite, (**b**) GO, (**c**) TRGO-100, and (**d**) TRGO-500 with high magnification micrograph displayed as insets to the figures, including the (**e**) digit width and (**f**) interspace distance of µ-SC (6).

**Figure 4 nanomaterials-11-01909-f004:**
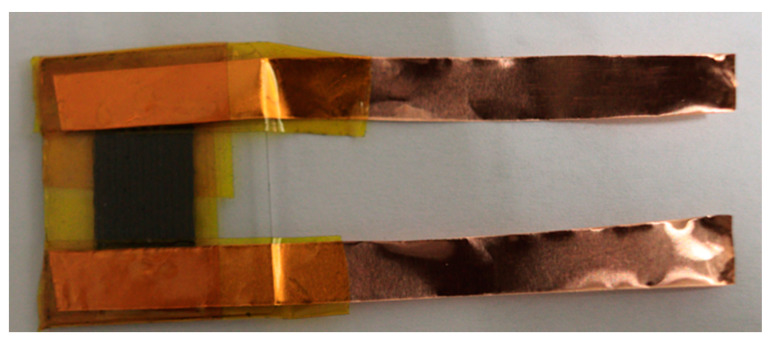
Digital photograph of TRGO µ-SC device.

**Figure 5 nanomaterials-11-01909-f005:**
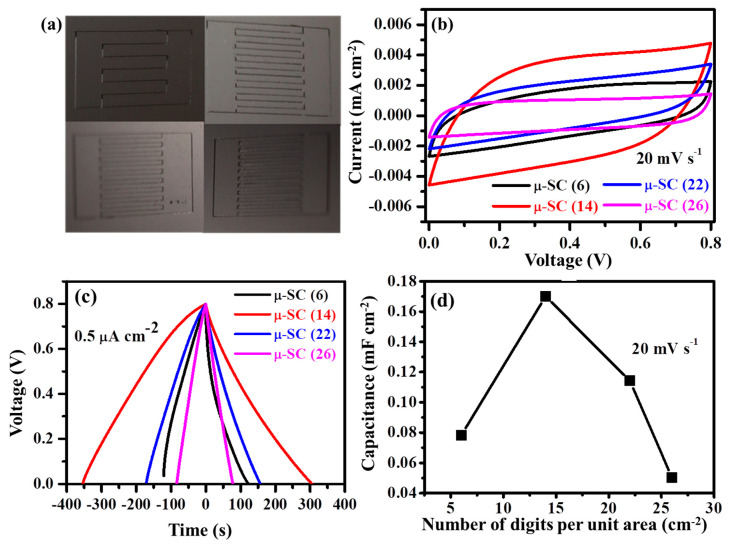
(**a**) Digital photographs; (**b**) CV curve at 20 mVs^−1^; (**c**) GCD curve at 0.5 µA cm^−2^ of TRGO µ-SC patterns with 6, 14, 22, and 26 cm^−2^; and (**d**) areal capacitances versus number of digits per unit area of the TRGO-300.

**Figure 6 nanomaterials-11-01909-f006:**
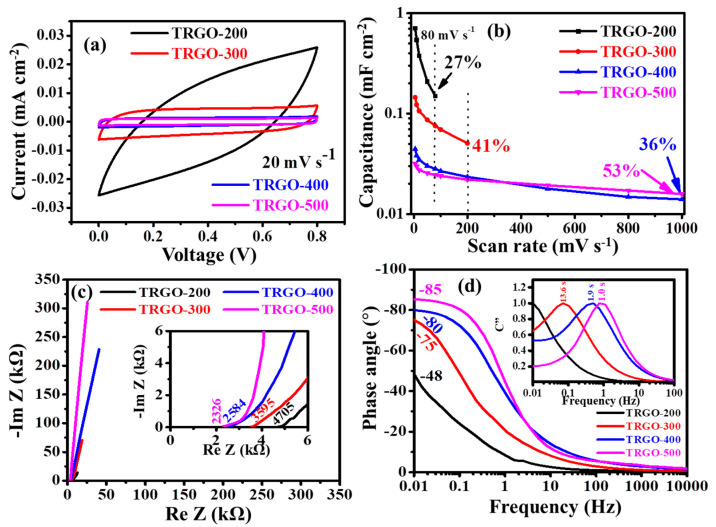
(**a**) CV curves at a scan rate of 20 mVs^−1^, (**b**) areal capacitance at various scan rates, (**c**) nyquist plot with the inset showing a magnified view of the high-frequency region and (**d**) phase angle as a function of frequency (inset: normalized imaginary capacitance (C″) as a function of frequency calculated from the EIS) for µ-SC (14) at different RT.

**Figure 7 nanomaterials-11-01909-f007:**
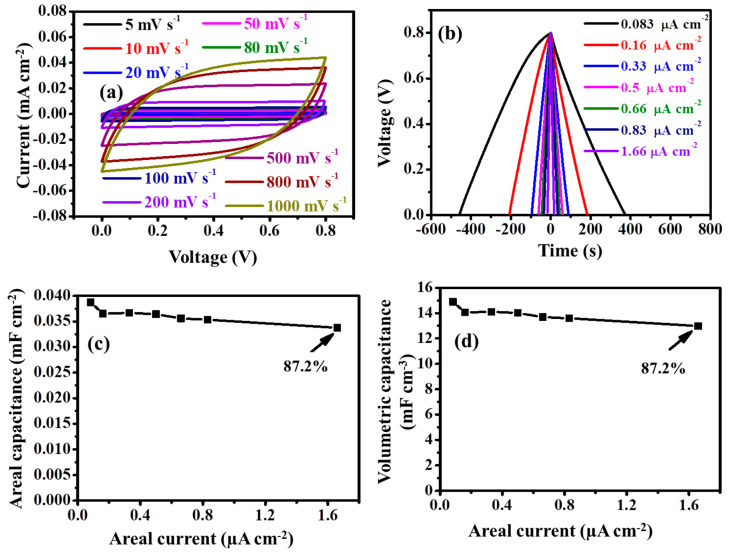
(**a**) CV curves at various scan rates, (**b**) GCD curves at various current density, (**c**) areal capacitance, and (**d**) volumetric capacitance as a function of current density for the TRGO-500 µ-SC (14).

**Figure 8 nanomaterials-11-01909-f008:**
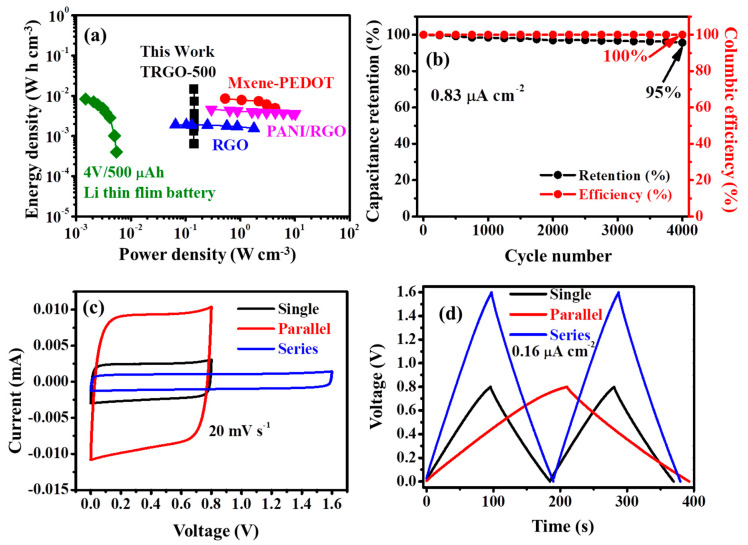
(**a**) Ragone plot of the TRGO-500 µ-SC compared to other similar devices and (**b**) capacitance retention and columbic efficiency of TRGO-500 versus cycling number at current density of 0.83 µA cm^−2^. The TRGO-500 (**c**) CV and (**d**) GCD curves of a single µ-SC, two µ-SC connected in series and parallel at a scan rate of 20 mV s^−1^, and a current density of 0.16 µA cm^−2^, respectively.

## Data Availability

Data is contained within the article or Appendix A.

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
