# Peer review of "Novel Thermally Reduced Graphene Oxide Microsupercapacitor Fabricated via Mask—Free AxiDraw Direct Writing"

_nanomaterials, 2021, doi:10.3390/nano11081909_

Round 1

Reviewer 1 Report

The submitted manuscript describes a μ-SC fabricating method based on a musk-free Axidraw apparatus, in which GO films are reduced on the MSG substrate. The authors showed that the obtained μ-SCs exhibited interesting electrochemical performance by tuning reducing temperature, number of digits per unity area, and connection configuration. From this point of view, the work may be interesting, but there are some obvious drawbacks that should be improved.

  1. There are some formatting, character, and language errors in the manuscript. The authors need to carefully review and improve the manuscript.
  2. Why are there no experiments with the heating temperature above 500 ℃, so it is not enough to prove that 500 ℃ is the optimal temperature.
  3. The abscissa label of the ATR-FTIR spectra is not correct.
  4. In Fig.3, the SEM images are too blurry to read.
  5. In Fig. 3, 5, 6, 7, the writing of decimal should be used correctly.
  6. In Reference, lots of names are wrong, such as names in Reference 2, 37...

Author Response

Reviewer 1 Comments

The submitted manuscript describes a μ-SC fabricating method based on a mask-free Axidraw apparatus, in which GO films are reduced on the MSG substrate. The authors showed that the obtained μ-SCs exhibited interesting electrochemical performance by tuning reducing temperature, number of digits per unity area, and connection configuration. From this point of view, the work may be interesting, but there are some obvious drawbacks that should be improved.

There are some formatting, character, and language errors in the manuscript. The authors need to carefully review and improve the manuscript.

Response:

We thank the reviewers for raising this comment; the authors have carefully reviewed the manuscript for better formatting and language editing .

Why are there no experiments with the heating temperature above 500 ℃, so it is not enough to prove that 500 ℃ is the optimal temperature.

Response:

Authors appreciate the reviewer’s comment. Experiments were also done with a heating temperature above 500 ºC. However, above this reducing temperature the MSG melted/deformed and the material also peeled off the surface of the MSG as mentioned under experimental section 2.4.

The abscissa label of the ATR-FTIR spectra is not correct.

Response:

The abscissa label of the ATR-FTIR spectra has been corrected.

In Fig.3, the SEM images are too blurry to read.

Response:

The SEM images have been modified on the manuscript for a better visibility

In Fig. 3, 5, 6, 7, the writing of decimal should be used correctly.

Response:

The writing of decimal has been corrected as pointed out by the reviewer.

In Reference, lots of names are wrong, such as names in Reference 2, 37...

Response:

All the references were checked for errors.

Reviewer 2 Report

This paper could be accepted after addressing the following questions:

(1) The introduction part is long but quite de-focus. It is strongly suggested to focus more on the difference of this work with others.

(2) Please proofread carefully, many typos were found. Such as "Is is well know..."; "This behaviour suggest that..."; "higher digits leads to..."; "ref [23]"...

(3) Please use "." to replace "," in Figures as it is difficult to understand.

(4) As for me, Figure 3(b-d) shows little information, please use higher magnification images.

(5) "The normalised to [0;1] CV curves..." could not be understood.

(6) In the conclusion part, please clearly state how the temperature and number of digits affect the performance of the micro supercapacitor.

(7) Please explain why TRGO-500 possesses better retention.

(8) Please explain why "The electric field between end and edge could lead to short". As for my understanding, they should possess the same potential.

(9) It is suggested to cite more references, such as "Polymers 202113(13), 2137; https://doi.org/10.3390/polym13132137". 

Author Response

Reviewer 2 comments

This paper could be accepted after addressing the following questions:

The introduction part is long but quite de-focus. It is strongly suggested to focus more on the difference of this work with others.

Response:

The introduction was revised and trimmed in accordance with the reviewer’s comment.

(2) Please proofread carefully, many typos were found. Such as "Is is well know..."; "This behaviour suggest that..."; "higher digits leads to..."; "ref [23]"...

Response:

The authors would like to thank the reviewers for this comment; the manuscript was proofread in order to correct the typos.

(3) Please use "." to replace "," in Figures as it is difficult to understand.

Response:

In all figures we have replaced the "," by ".".

(4) As for me, Figure 3(b-d) shows little information, please use higher magnification images.

Response:

The authors would like to thank the review for this comment. The high magnification images have been added as insets in Fig. 3(a-d).

(5) "The normalised to [0;1] CV curves..." could not be understood.

Response:

We agree with the reviewer and hence the Figure has been removed

(6) In the conclusion part, please clearly state how the temperature and number of digits affect the performance of the micro supercapacitor.

Response:

The authors have clearly stated how the temperature and number of digits affected the performance of the microsupercapacitors in the conclusions as suggested by the revewer.

(7) Please explain why TRGO-500 possesses better retention.

Response:

Higher reducing temperature has been reported to increase the carbon content. The low temperature reduced TRGO is rich in resistive oxygen which does not allow the capacitor to be fully charge at high scan rate unlike the high temperature reduced TRGO-500[1,2].

[1] Zhao, B.; Liu, P.; Jiang, Y.; Pan, D.; Tao, H.; Song, J.; Fang, T.; Xu, W. Supercapacitor performances of thermally reduced graphene oxide. J. Power Sources 2012, 198, 423–427, doi:10.1016/j.jpowsour.2011.09.074.

[2]Claramunt, S.; Varea, A.; Lópezlópez-Díaz, D.; Mercedes Velázquezvelázquez, M.; Cornet, A.; Cirera, A. The Importance of Interbands on the Interpretation of the Raman Spectrum of Graphene Oxide. J. Phys. Chem. C 2015, 119, 10123–10129, doi:10.1021/acs.jpcc.5b01590.

(8) Please explain why "The electric field between end and edge could lead to short". As for my understanding, they should possess the same potential.

Response:

This is true for parallel capacitors. According to classical electrodynamics and the electric field simulations by Hota at el. [3] on fractional µ-SC electrode design showed that the edge intensive electrode design has an increase in electric field which also leads to a high capacitance. In our work, the electric field might have increased in such a manner that charge flows or leaks (short circuits) from the negative to the positive via the electrolyte [4]. This has been rewritten in the revised manuscript.

[3] Hota, M.K.; Jiang, Q.; Mashraei, Y.; Salama, K.N.; Alshareef, H.N. Fractal Electrochemical Microsupercapacitors. Adv. Electron. Mater. 2017, 3, 1–9, doi:10.1002/aelm.201700185.

[4] Liu, N.; Gao, Y. Recent Progress in Micro-Supercapacitors with In-Plane Interdigital Electrode Architecture. Small 2017, 13, doi:10.1002/smll.201701989.

(9) It is suggested to cite more references, such as "Polymers 202113(13), 2137; https://doi.org/10.3390/polym13132137".

Response:

The authors have cited more reference as suggest by the reviewer.

Reviewer 3 Report

In this manuscript by Maphiri et al, the authors fabricate all-solid-state thermally reduced Graphene Oxide (TRGO) microsupercapacitors (μ-SCs) by mask-free axidraw sketching apparatus. The electrochemical performances of μ-SCs under different annealing temperature were evaluated and compared. However, there are so limited structure analysis for the TRGO materials. The following issues must be carefully concerned before it can be pulished in Nanomaterials.

  1. SEM and AFM images to show lateral size and thickness of GO should be presented in the revision.
  2. The mass loading of TRGO on specific surface of MSG should be determined and the mass specific capacitance of it under different reducing temperature should be provided.
  3. There are many English errors, for example, "be hummer’s method [5,10–18] which is safest and efficient compared", "which is very different to the masked technique", and "a musk-free Axidraw apparatus", etc. 

Author Response

Reviewer 3

In this manuscript by Maphiri et al, the authors fabricate all-solid-state thermally reduced Graphene Oxide (TRGO) microsupercapacitors (μ-SCs) by mask-free axidraw sketching apparatus. The electrochemical performances of μ-SCs under different annealing temperature were evaluated and compared. However, there are so limited structure analysis for the TRGO materials. The following issues must be carefully concerned before it can be pulished in Nanomaterials.

SEM and AFM images to show lateral size and thickness of GO should be presented in the revision.

Response:

Authors appreciate the reviewer for raising this concern. The SEM and AFM images of GO won’t increase the novelty of this since GO is not the target. However, the SEM and AFM images showing the lateral size and the thickness of the GO is presented on the below for the reviewer which was not included in the main text as we don’t find it adding any value to the work. The SEM shows that GO shows a lateral size of around 2 µm while the AFM shows the thickness of around 1.5 nm corresponding to around 4 layers of graphene.

Fig. 2 (a) SEM micrograph, AFM (b) image and (c) height profile

The mass loading of TRGO on specific surface of MSG should be determined and the mass specific capacitance of it under different reducing temperature should be provided.

Response:

We thank the reviewer for such a comment aimed at improving the nature of the manuscript. The authors think that determining the mass specific capacitance would be very difficult, since the reduced and µ-SC patterned TRGO on MSG (see Fig. S 1(e), supplementary information) was cut into individual µ-SC then followed by the scraping of the unwanted TRGO on the outside of the µ-SC patterns. Thus, the mass of the cut MSG is unknown and make it impossible to also determine the mass of the µ-SC TRGO active material.

There are many English errors, for example, "be hummer’s method [5,10–18] which is safest and efficient compared", "which is very different to the masked technique", and "a musk-free Axidraw apparatus", etc. 

Response:

We thank the review for such a fruitful comment; the manuscript was proof-read in order to change the English errors and typos.

Round 2

Reviewer 1 Report

I think the authors have already addressed my comments, but the English polishing is needed.

Reviewer 2 Report

Can be accepted in the present form

Reviewer 3 Report

After revision, most of previous issues are concerned. I can suggest its publication.